# Bandlimiting Neural Networks Against Adversarial Attacks

## Abstract

In this paper, we study the adversarial attack and defence problem in deep learning from the perspective of Fourier analysis. We first explicitly compute the Fourier transform of deep ReLU neural networks and show that there exist decaying but non-zero high frequency components in the Fourier spectrum of neural networks. We then demonstrate that the vulnerability of neural networks towards adversarial samples can be attributed to these insignificant but non-zero high frequency components. Based on this analysis, we propose to use a simple *post-averaging* technique to smooth out these high frequency components to improve the robustness of neural networks against adversarial attacks. Experimental results on the ImageNet and the CIFAR-10 datasets have shown that our proposed method is universally effective to defend many existing adversarial attacking methods proposed in the literature, including FGSM, PGD, DeepFool and C&W attacks. Our post-averaging method is simple since it does not require any re-training, and meanwhile it can successfully defend over 80-96% of the adversarial samples generated by these methods without introducing significant performance degradation (less than 2%) on the original clean images.

## 1 Introduction

Although deep neural networks (DNN) have shown to be powerful in many machine learning tasks, Szegedy et al. (2013) found that they are vulnerable to *adversarial samples*. Adversarial samples are subtly altered inputs that can fool the trained model to produce erroneous outputs. They are more commonly seen in image classification task and typically the perturbations to the original images are so small that they are imperceptible to human eye.

Research in adversarial attacks and defences is highly active in recent years. In the attack side, many attacking methods have been proposed (Szegedy et al., 2013; Goodfellow et al., 2014; Papernot et al., 2016a; Papernot et al., 2017; Moosavi-Dezfooli et al., 2016; Kurakin et al., 2016; Madry et al., 2017; Carlini and Wagner, 2017a; Chen et al., 2017; Alzantot et al., 2018; Brendel et al., 2017), with various ways to generate effective adversarial samples to circumvent new proposed defence methods. However, since different attacks usually are effective to different defences or datasets, there is no consensus on which attack is the strongest. Hence for the sake of simplicity, in this work, we will evaluate our proposed defence approach against four popular attacks for empirical analysis. In the defence side, various defence mechanisms have also been proposed, including adversarial training (Rozsa et al., 2016; Kurakin et al., 2016; Tramèr et al., 2017; Madry et al., 2017), network distillation (Papernot et al., 2016b), gradient masking (Nguyen and Sinha, 2017), adversarial detection (Feinman et al., 2017) and adding modifications to neural networks (Xie et al., 2017). Nonetheless, many of them were quickly defeated by new types of attacks (Carlini and Wagner, 2016; 2017b;c;a; Athalye et al., 2018; Athalye and Carlini, 2018; Alzantot et al., 2018). Madry et al. (2017) tried to provide a theoretical security guarantee for adversarial training by a min-max loss formulation, but the difficulties in non-convex optimization and in finding the ultimate adversarial samples for training may loosen this robustness guarantee. As a result, so far there is no defence that is universally robust to all adversarial attacks.

Along the line of researches, there were also investigations into the properties and existence of adversarial samples. Szegedy et al. (2013) first observed the transferability of adversarial samples across models trained with different hyper-parameters and across different training sets. They also

attributed the adversarial samples to the low-probability blind spots in the manifold. In (Goodfellow et al., 2014), the authors explained adversarial samples as "a result of models being too linear, rather than too nonlinear." In (Papernot et al., 2016), the authors showed the transferability occurs across models with different structures and even different machine learning techniques in addition to neural networks. In summary, the general existence and transferability of adversarial samples are well known but the reason of adversarial vulnerability still needs further investigation.

Generally speaking, when we view neural network as a multivariate function $f(\mathbf{x})$ of input $\mathbf{x}$, if a small imperceptible perturbation $\Delta\mathbf{x}$ leads to a huge fluctuation $\Delta f(\mathbf{x})$, the large quantity $\Delta f(\mathbf{x})/\Delta\mathbf{x}$ essentially corresponds to high frequency components in the Fourier spectrum of $f(\mathbf{x})$. In this paper, we will start with the Fourier analysis of neural networks and elucidate why there always exist some decaying but nonzero high frequency response components in neural networks. Based on this analysis, we show that neural networks are inherently vulnerable to adversarial samples due to the underlying model structure. Next, we propose a simple *post-averaging* method to tackle this problem. Our proposed method is fairly simple since it works as a post-processing stage of any given neural network models and it does not require re-training the networks at all. Furthermore, we have evaluated the post-averaging method against four popular adversarial attacking methods and our method is shown to be universally effective in defending all examined attacks. Experimental results on the ImageNet and the CIFAR-10 datasets have shown that our simple post-averaging method can successfully defend over 80-96% of the adversarial samples generated by these attacks with little performance degradation (less than 2%) on the original clean images.

## 2 FOURIER ANALYSIS OF NEURAL NETWORKS

In order to understand the behaviour of adversarial samples, it is essential to find the Fourier transform of neural networks. Fortunately, for some widely used neural networks, namely fully-connected neural networks using ReLU activation functions, we may explicitly derive their Fourier transform under some minor conditions. As we will show, these theoretical results will shed light on how adversarial samples happen in neural networks.

### 2.1 FOURIER TRANSFORM OF FULLY-CONNECTED ReLU NEURAL NETWORKS

As we know, any fully-connected ReLU neural networks (prior to the softmax layer) essentially form piece-wise linear functions in input space. Due to space limit, we will only present the main results in this section and the proofs and more details may be found in Appendix.

**Definition 2.1.** *A piece-wise linear function is a continuous function $f : \mathbb{R}^n \to \mathbb{R}$ such that there are some hyperplanes passing through origin and dividing $\mathbb{R}^n$ into $M$ pairwise disjoint regions $\mathfrak{R}_m$, $(m = 1, 2, ..., M)$, on each of which $f$ is linear:*

$$f(\mathbf{x}) = \begin{cases} \mathbf{w}_1 \cdot \mathbf{x} & \mathbf{x} \in \mathfrak{R}_1 \\ \mathbf{w}_2 \cdot \mathbf{x} & \mathbf{x} \in \mathfrak{R}_2 \\ \vdots \\ \mathbf{w}_M \cdot \mathbf{x} & \mathbf{x} \in \mathfrak{R}_M \end{cases}$$

**Lemma 2.2.** *Composition of a piece-wise linear function with a ReLU activation function is also a piece-wise linear function.*

**Theorem 2.3.** *The output of any hidden unit in an unbiased fully-connected ReLU neural network is a piece-wise linear function.*

This is straightforward because the input to any hidden node is a linear combination of piece-wise linear functions and this input is composed with the ReLU activation function to yield the output, which is also piece-wise linear. However, each region $\mathfrak{R}_m$ is the intersection of a different number of half-spaces, enclosed by various hyperplanes in $\mathbb{R}^n$. In general, these regions $\mathfrak{R}_m$ $(m = 1, \cdots, M)$ do not have simple shapes. For the purpose of mathematical analysis, we need to decompose each region into a union of some well-defined shapes having a uniform form, which is called *infinite simplex*.

**Definition 2.4.** *Let* $\mathbf{V} = \{\mathbf{v}_1, \mathbf{v}_2, ..., \mathbf{v}_n\}$ *be a set of* $n$ *linearly independent vectors in* $\mathbb{R}^n$. *An infinite simplex,* $\mathfrak{R}_{\mathbf{V}}^+$, *is defined as the region linearly spanned by* $\mathbf{V}$ *using only positive weights:*

$$\mathfrak{R}_{\mathbf{V}}^+ = \left\{ \sum_{k=1}^n \alpha_k \mathbf{v}_k \ \middle| \ \alpha_k > 0, \ \ k = 1, 2, \cdots, n \right\} \tag{1}$$

**Theorem 2.5.** *Each piece-wise linear function* $f(\mathbf{x})$ *can be formulated as a summation of some simpler functions:* $f(\mathbf{x}) = \sum_{l=1}^L f_l(\mathbf{x})$, *each of which is linear and non-zero only in an infinite simplex as follows:*

$$f_l(\mathbf{x}) = \begin{cases} \mathbf{w}_l \cdot \mathbf{x} & \mathbf{x} \in \mathfrak{R}_{\mathbf{V}_l}^+ \\ 0 & otherwise \end{cases} \tag{2}$$

*where* $\mathbf{V}_l$ *is a set of* $n$ *linearly independent vectors, and* $\mathbf{w}_l$ *is a weight vector.*

In practice, we can always assume that the input to neural networks, $\mathbf{x}$, is bounded. As a result, for computational convenience, we may normalize all inputs $\mathbf{x}$ into the unit hyper-cube, $U_n = [0, 1]^n$. Obviously, this assumption can be easily incorporated into the above analysis by multiplying each $f_l(\mathbf{x})$ in eq.(2) by $\prod_{r=1}^n h(x_r)h(1 - x_r)$ where $h(x)$ is the Heaviside step function. Alternatively, we may simplify this term by adding $n^2$ additional hyperplanes to further split the input space to ensure all the elements of $\mathbf{x}$ do not change signs within each region $\mathfrak{R}_{\mathbf{V}_q}^+$. In this case, within each region $\mathfrak{R}_{\mathbf{V}_q}^+$, the largest absolute value among all elements of $\mathbf{x}$ is always achieved by a specific element, which is denoted as $r_q$. In other words, the dimension $x_{r_q}$ achieves the largest absolute value inside $\mathfrak{R}_{\mathbf{V}_q}^+$. Similarly, the normalized piece-wise linear function may be represented as a summation of some functions: $f(\mathbf{x}) = \sum_{q=1}^Q g_q(\mathbf{x})$, where each $g_q(\mathbf{x})$ $(q = 1, 2, \cdots, Q)$ has the following form:

$$g_q(\mathbf{x}) = \begin{cases} \mathbf{w}_q \cdot \mathbf{x} \, h(1 - x_{r_q}) & \mathbf{x} \in \mathfrak{R}_{\mathbf{V}_q}^+ \\ 0 & otherwise \end{cases}$$

For every $\mathbf{V}_q$, there exists an $n \times n$ invertible matrix $\mathbf{A}_q$ to linearly transform all vectors of $\mathbf{V}_q$ into standard basis vectors $\mathbf{e}_i$ in $\mathbb{R}^n$. As a result, each function $g_q(\mathbf{x})$ may be represented in terms of standard bases $\mathbf{V}_* = \{\mathbf{e}_1, \cdots, \mathbf{e}_n\}$ as follows:

$$g_q(\mathbf{x}) = \begin{cases} \bar{\mathbf{w}}_q \cdot \bar{\mathbf{x}}_q \, h(1 - \mathbf{1} \cdot \bar{\mathbf{x}}_q) & \bar{\mathbf{x}}_q \in \mathfrak{R}_{\mathbf{V}_*}^+ \\ 0 & otherwise \end{cases}$$

where $\bar{\mathbf{x}}_q = \mathbf{x}\mathbf{A}_q^T$, and $\bar{\mathbf{w}}_q = \mathbf{w}_q\mathbf{A}_q^{-1}$.

**Lemma 2.6.** *Fourier transform of the following function:*

$$s(\mathbf{x}) = \begin{cases} h(1 - \mathbf{1} \cdot \mathbf{x}) & \mathbf{x} \in \mathfrak{R}_{\mathbf{V}_*}^+ \\ 0 & otherwise \end{cases}$$

*may be presented as:*

$$S(\boldsymbol{\omega}) = \left( \frac{-\mathbf{i}}{\sqrt{2\pi}} \right)^n \sum_{r=0}^n \frac{e^{-\mathbf{i}\omega_r}}{\prod_{r' \neq r} (\omega_{r'} - \omega_r)} \tag{3}$$

*where* $\omega_r$ *is the* $r$-*th component of frequency vector* $\boldsymbol{\omega}$ $(r = 1, \cdots, n)$, *and* $\omega_0 = 0$.

Finally we derive the Fourier transform of fully-connected ReLU neural networks as follows.

**Theorem 2.7.** *The Fourier transform of the output of any hidden node in a fully-connected unbiased[1] ReLU neural network may be represented as* $\sum_{q=1}^Q \mathbf{w}_q \mathbf{A}_q^{-1} \nabla S(\boldsymbol{\omega}\mathbf{A}_q^{-1})$, *where* $\nabla$ *denote the differential operator.*

---

[1] For mathematical convenience, we assume neural networks have no biases here. However, regular neural networks with biases may be reformulated as unbiased ones by adding another dimension of constants. Thus, the main results here are equally applicable to both cases. Note that regular neural networks with biases are used in our experiments in this paper.

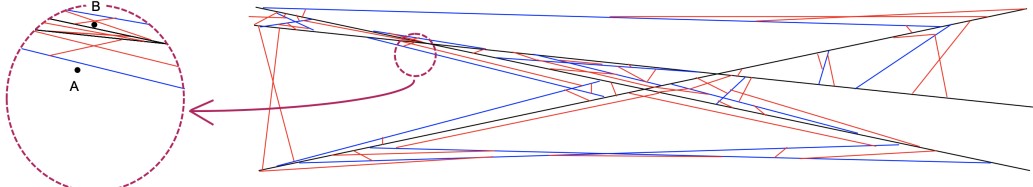

Figure 1: Illustration of input space divided into sub-regions by a biased neural network. The black lines are the hyperplanes for the first network layer, while the blue lines are for the second layer and the red lines are for the third layer. A small perturbation from point $A$ to point $B$ may possibly cross many hyperplanes.

Obviously, neural networks are the so-called approximated bandlimited models as defined in (Jiang, 2019), which have decaying high frequency components in Fourier spectrum. Theorem 2.7 further suggests that the matrices $\mathbf{A}_q^{-1}$ may contribute to the high frequency components when the corresponding region $\mathfrak{R}_{\mathbf{V}_q}^+$ are too small. This is clear because the determinant of $\mathbf{A}_q$ is proportional to the volume of $\mathfrak{R}_{\mathbf{V}_q}^+$ in $\mathbb{R}^n$. In summary, the high frequency components of neural networks are mostly attributed to these tiny regions in the input space. As we will show later, these small regions may be explicitly exploited to generate adversarial samples for neural networks.

## 2.2 UNDERSTANDING ADVERSARIAL SAMPLES

As shown in Theorem 2.3, neural network may be viewed as a sequential division of the input space into many small regions, as illustrated in Figure 1. Each layer is a further division of the existing regions from the previous layers, with each region being divided differently. Hence a neural network with multiple layers would result in a tremendous amount of sub-regions in the input space. For example, when cutting an $n$-dimensional space using $N$ hyperplanes, the maximum number of regions may be computed as $\binom{N}{0} + \binom{N}{1} + \cdots + \binom{N}{n}$. For a hidden layer of $N = 1000$ nodes and input dimension is $n = 200$, the maximum number of regions is roughly equal to $10^{200}$. In other words, even a middle-sized neural network can partition input space into a huge number of sub-regions, which can easily exceed the total number of atoms in the universe. When we learn a neural network, we can not expect there is at least one training sample inside each region. For those regions that do not have any training sample, the resultant linear functions in them may be arbitrary since they do not contribute to the training objective function at all. Of course, most of these regions are extremely small in size. When we measure the expected loss function over the entire space, their contributions are negligible since the chance for a randomly sampled point to fall into these tiny regions is extremely small. However, adversarial attack is imposing a new challenge since adversarial samples are not naturally sampled. Given that the total number of regions is huge, those tiny regions are almost everywhere in the input space. For any data point in the input space, we almost surely can find such a tiny region in proximity where the linear function is arbitrary. If a point inside this tiny region is selected, the output of the neural network may be unexpected. We believe that these tiny unlearned regions may be a major reason why neural networks are vulnerable to adversarial samples.

In layered deep neural networks, the linear functions in all regions are not totally independent. If we use $\mathbf{v}^{(l)}$ to denote the weight matrix in layer $l$, the resultant linear weight $\mathbf{w}_k$ in eq.(2) is actually the sum of all concatenated $\mathbf{v}^{(l)}$ along all active paths. When we make a small perturbation $\Delta \mathbf{x}$ to any input $\mathbf{x}$, the fluctuation in the output of any hidden node can be approximated represented as:

$$\Delta f(\mathbf{x}) \propto N \cdot \prod_l \mathbf{E} \left[ |\mathbf{v}_{ij}^{(l)}| \right] \tag{4}$$

where $N$ denotes the total number of hyperplanes to be crossed when moving $\mathbf{x}$ to $\mathbf{x} + \Delta \mathbf{x}$. In any practical neural network, we normally have at least tens of thousands of hyperplanes crossing the hypercube $U_n = [0, 1]^n$. In other words, for any input $\mathbf{x}$ in a high-dimensional space, a small perturbation can always easily cross a large number of hyperplanes to enter a tiny unlearned region. When $N$ is fairly large, the above equation indicates that the output of a neural network can still fluctuate dramatically even after all weight vectors are regularized by $L_1$ or $L_2$ norm. As a reference,

we have verified this on some ImageNet data using a VGG16 model. When PGD is used to generate adversarial samples with average perturbation $||\Delta \mathbf{x}||_2 \le 0.35$, which is extremely small perturbation since $\mathbf{x}$ has over a hundred thousand dimensions on ImageNet, we have observed that in average about $N = 5278$ hyperplanes are crossed per layer even after such a small perturbation is added.

At last, since the ubiquitous existence of unlearned tiny regions is an intrinsic property of neural networks given its current model structure, we believe that adversarial training strategies will not be sufficient to completely get rid of adversarial samples. In principle, neural networks must be strictly bandlimited to filter out those decaying high frequency components in order to completely eliminate all adversarial samples. We definitely need more research efforts to figure out how to do this effectively and efficiently for neural networks.

## 3 THE PROPOSED DEFENCE APPROACH: POST-AVERAGING

### 3.1 POST-AVERAGING

In this paper, we propose a simple post-processing method to smooth out those high frequency components as much as possible, which relies on a simple idea similar to moving-average in one-dimensional sequential data. Instead of generating prediction merely from one data point, we use the averaged value within a small neighborhood around the data point, which is called *post-averaging* here. Mathematically, the post-averaging is computed as an integral over a small neighborhood centered at the input:

$$f_C(\mathbf{x}) = \frac{1}{\mathbf{V}_C} \int \cdots \int_{\mathbf{x}' \in C} f(\mathbf{x} - \mathbf{x}') \, d\mathbf{x}' \tag{5}$$

where $\mathbf{x}$ is the input and $f(\mathbf{x})$ represents the output of the neural network, and $C$ denotes a small neighborhood centered at the origin and $\mathbf{V}_C$ denotes its volume. When we choose $C$ to be an $n$-sphere in $\mathbb{R}^n$ of radius $r$, we may simply derive the Fourier transform of $f_C(\mathbf{x})$ as follows:

$$F_C(\boldsymbol{\omega}) = F(\boldsymbol{\omega}) \frac{1}{\mathbf{V}_C} \int \cdots \int_{\mathbf{x}' \in C} e^{-i\mathbf{x}' \cdot \boldsymbol{\omega}} \, d\mathbf{x}' = F(\boldsymbol{\omega}) \frac{\Gamma(\frac{n}{2} + 1)}{\pi^{\frac{n}{2}}} \frac{J_{\frac{n}{2}}(r|\boldsymbol{\omega}|)}{(r|\boldsymbol{\omega}|)^{\frac{n}{2}}} \tag{6}$$

where $J_{\frac{n}{2}}(\cdot)$ is the first kind Bessel function of order $n/2$. Since the Bessel functions, $J_\nu(\omega)$, decay with rate $1/\sqrt{\omega}$ as $|\boldsymbol{\omega}| \to \infty$ (Watson, 1995), we have $F_C(\boldsymbol{\omega}) \sim \frac{F(\boldsymbol{\omega})}{(r|\boldsymbol{\omega}|)^{\frac{n+1}{2}}}$ as $|\boldsymbol{\omega}| \to \infty$. Therefore, if $r$ is chosen properly, the post-averaging operation can significantly bandlimit neural networks by smoothing out high frequency components. Note that the similar ideas have been used in (Jiang et al., 1999; Jiang and Lee, 2003) to improve robustness in speech recognition.

### 3.2 SAMPLING METHODS

However, it is intractable to compute the above integral for any meaningful neural network used in practical applications. In this work, we propose to use a simple numerical method to approximate it. For any input $\mathbf{x}$, we select $K$ points in the neighborhood $C$ centered at $\mathbf{x}$, i.e. $\{\mathbf{x}_1, \mathbf{x}_2, \cdots, \mathbf{x}_K\}$, to approximately compute the integral as

$$f_C(\mathbf{x}) \approx \frac{1}{K} \sum_{k=1}^{K} f(\mathbf{x}_k). \tag{7}$$

Obviously, in order to defend against adversarial samples, it is important to have samples outside the current unlearned tiny region. In the following, we use a simple sampling method based on directional vectors. To generate a relatively even set of samples for eq.(7), we first determine some directional vectors $\hat{\mathbf{v}}$, and then move the input $\mathbf{x}$ along these directions using several step sizes within the sphere of radius $r$:

$$\mathbf{x}' = \mathbf{x} + \lambda \cdot \hat{\mathbf{v}} \tag{8}$$

where $\lambda = [\pm \frac{r}{3}, \pm \frac{2r}{3}, \pm r]$, and $\hat{\mathbf{v}}$ is a selected unit-length directional vector. For each selected direction, we generate six samples within C along both the positive and the negative directions to ensure efficiency and even sampling. We use this implementation for the convenience to extend with different types of sampling strategies.

We tried several direction sampling strategies, including using the directions towards the closest region boundaries, and found that the simple *random direction sampling* gives the best performance. In this sampling method, we fill the directional vectors with random numbers generated from a standard normal distribution, and then normalize them to have unit length.

## 4 EXPERIMENTS

In this section, we evaluate the above post-averaging method on defending against several popular adversarial attacking methods.

### 4.1 EXPERIMENTAL SETUP

- **Dataset**: We evaluated our method on both the ImageNet (Russakovsky et al., 2015) and CIFAR-10 (Krizhevsky et al., 2009) datasets. Since our proposed post-averaging method does not need to re-train neural networks, we do not need to use any training data in our experiments.

  For evaluation purpose, we use the validation set of the ImageNet dataset. The validation set consists of 50000 images labelled into 1000 categories. For computational efficiency, we randomly choose 5000 images from the ImageNet validation set and evaluate our model on these 5000 images.

  For the CIFAR-10 dataset, we use the full test set, which consists of 10000 images labelled into 10 categories.

- **Target model**: For model on ImageNet, we use a pre-trained ResNet-152 (He et al., 2016) network that is available from PyTorch, while for CIFAR-10, we use a pre-trained ResNet-110 network from Yerlan Idelbayev [2]. In our experiments, we directly use these pre-trained models without any modification.

- **Source of adversarial attacking methods**: We use Foolbox (Rauber et al., 2017), an open source tool box to generate adversarial samples using different adversarial attacking methods. In this work, we tested our method against four popular attacking methods in the literature: Fast Gradient Sign method (FGSM) (Goodfellow et al., 2014), Projected Gradient Descent (PGD) method (Kurakin et al., 2016; Madry et al., 2017), DeepFool (DF) attack method (Moosavi-Dezfooli et al., 2016) and Carlini & Wagner (C&W) L2 attack method (Carlini and Wagner, 2017a). We used these attack methods in their default settings.

- **Threat model**: In our experiments, we use an $l_\infty$ norm $\epsilon$ to constrain the allowed perturbation distance.

### 4.2 EVALUATION CRITERIA

For each experiment, we define:

- **Clean set**: The dataset that consists of the original images from ImageNet or CIFAR-10.
- **Attacked set**: For every correctly classified image in the Clean set, if an adversarial sample is successfully generated under the attacking criteria, the original sample is replaced with the adversarial sample; if no adversarial sample is found, the original sample is kept in the dataset. Meanwhile, all the misclassified images are kept in the dataset without any change. Therefore the Attacked set has the same number of images as the clean set.

In our experiments, we evaluate the original network and the network defended by post-averaging on both the Clean and the Attacked sets. The performance is measured in terms of :

- **Accuracy**: number of correctly classified images over the whole dataset.
- **Defence rate**: number of successfully defended adversarial samples over the total number of adversarial samples in the Attacked set. By "successfully defended", it refers to the case where an adversarial sample is correctly classified after the original model is defended by the post-averaging approach.

---

[2] https://github.com/akamaster/pytorch_resnet_cifar10/tree/master/pretrained_models

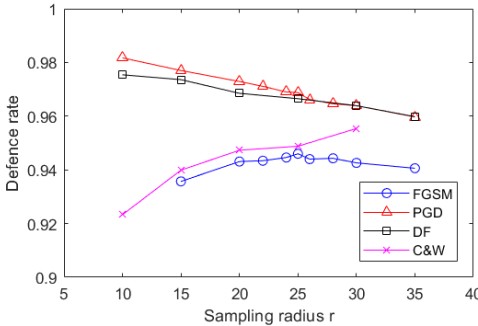

Figure 2: Defence rate of post-averaging when using different sampling radius $r$ on ImageNet.

Table 1: Performance of post-averaging defending against different attacking methods ($\epsilon = {}^8/_{255}$, and $K = 15$).

| attack, defence | Dataset | Original Model Top-1 Accuracy Clean | Attacked | Defended by Post-Averaging Top-1 Accuracy Clean | Attacked | Defence Rate | #Adv |
|---|---|---|---|---|---|---|---|
| FGSM, random(r=30) |  |  | 0.0750 | 0.7734 | 0.7488 | 0.9426 | 3500 |
| PGD, random(r=30) | ImageNet | 0.7750 | 0.0004 | 0.7732 | 0.7606 | 0.9639 | 3873 |
| DF, random(r=30) |  |  | 0.0350 | 0.7730 | 0.7614 | 0.9639 | 3710 |
| C&W, random(r=30) |  |  | 0.0090 | 0.7734 | 0.7548 | 0.9554 | 3830 |
| FGSM, random(r=6) |  |  | 0.1816 | 0.9247 | 0.8022 | 0.8080 | 7552 |
| PGD, random(r=6) | CIFAR-10 | 0.9368 | 0.0000 | 0.9255 | 0.8841 | 0.9330 | 9368 |
| DF, random(r=6) |  |  | 0.1968 | 0.9254 | 0.8626 | 0.8872 | 7413 |
| C&W, random(r=6) |  |  | 0.0322 | 0.9257 | 0.8902 | 0.9367 | 9046 |

## 4.3 EXPERIMENTAL RESULTS

Table 1 shows the performance of our defence approach against different attacking methods. In this table, the samples for post-averaging are selected within an $n$-sphere of radius $r$ as in eq.(8), with $K = 15$ different directions. Thus results in a total of $15 \times 2 \times 3 + 1 = 91$ samples (including the input) for each input image to be used in eq.(7). Moreover, all the adversarial samples generated are restricted to be within the perturbation range $\epsilon = {}^8/_{255}$. We show the top-1 accuracy of the original model and the defended model on both the Clean and the Attacked set respectively, as well as the defence rate of the defended model. Besides, we also show the number of adversarial samples successfully generated by each attacking method in the last column.

From Table 1, we can see that our proposed defence approach is universally robust to all of the attacking methods we have examined. It has achieved above 80-96% defence rates in all the experiments with only a minor performance degradation in the Clean set (less than 2%). Especially on the ImageNet dataset, our method is able to defend about 95% of the adversarial samples. However, an interesting observation from the experimental results is that the defence rate in the CIFAR-10 dataset is lower than the usually more challenging ImageNet dataset. We think this may be because data points are sparser in the ImageNet space than in the CIFAR-10 space, as ImageNet has a much larger dimensionality.

Generally, using a larger sampling radius $r$ can increase the chance of moving out of the unlearned regions as we desired, but it will also introduce more noise that can harm the prediction accuracy; On the other hand, using a smaller sampling radius $r$ can reduce the performance degradation but it may not be sufficient to defend against adversarial samples. The optimal value for $r$ varies with different datasets due to their dimensionality and data sparsity. In experiments, we found that $r = 30$ for ImageNet and $r = 6$ for CIFAR-10 achieved relatively better performance. Figure 2 shows how the model defence rate on ImageNet varies with different $r$. As shown in the figure, the optimal value for $r$ also varies in different attacking methods, but the performance variations are small. In general, our model retains high defence rate throughout the $r$ range $[15, 30]$.

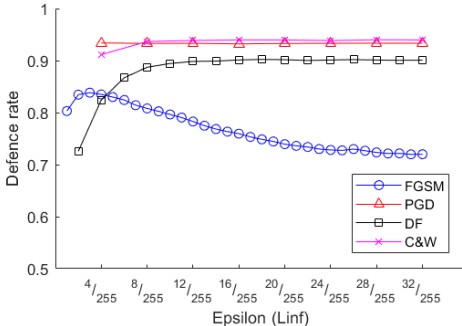

Figure 3: Defence rate of post-averaging on different allowed perturbation distances $\epsilon$ on CIFAR-10.

Table 2: Performance of post-averaging on ImageNet with different number of sampling directions ($\epsilon = {}^8/_{255}$ and $r = 30$).

| | Original Model | | Defended by Post-Averaging | | | | Averaged |
|---|---|---|---|---|---|---|---|
| | Top-1 Accuracy | | Top-1 Accuracy | | Defence | | Inference |
| attack, defence | Clean | Attacked | Clean | Attacked | Rate | #Adv | Time |
| PGD, random(K=6) | | 0.0004 | 0.7736 | 0.7606 | 0.9636 | 3873 | 0.18s |
| PGD, random(K=15) | | 0.0004 | 0.7730 | 0.7604 | 0.9636 | 3873 | 0.40s |
| PGD, random(K=30) | 0.7750 | 0.0006 | 0.7732 | 0.7614 | 0.9644 | 3874 | 0.86s |
| PGD, random(K=60) | | 0.0004 | 0.7734 | 0.7618 | 0.9649 | 3874 | 1.74s |

We also tested the effect of $K$, the number of sampling directions used, on the model performance. From Table 2, we can see that our model performance is not very sensitive to $K$. It is able to achieve a good defence rate with only $K = 6$, that is, 37 samples used for each input image. In implementation, these samples can be easily packed into a mini-batch for fast computation in GPUs. When running on the same machine, we measured the averaged inference time for a single input image on the original network as $0.04$ seconds, while the inference time for our models with different $K$ are shown in Table 2. By comparison, we can know that the inference time after adding post-averaging is roughly $\frac{2}{3}K$ of the original inference time.

At last, we evaluated our post-averaging defence approach against attacks with different allowed perturbation ranges $\epsilon$. The results are shown in Figure 3. As we can see, our model retains very good attack defence rate up to $\epsilon = {}^{32}/_{255}$. Note that the defence rate against PGD and C&W doesn't change much along the variation of $\epsilon$, this is because PGD and C&W have already successfully generated adversarial samples for most of the correctly classified inputs when $\epsilon$ is small. Hence their generated adversarial samples will not change much when using larger $\epsilon$. For FGSM, our method yields lower defending performance. The possible reason is that FGSM tends to generate much larger perturbations than other three stronger attacking methods under the same setting. A large perturbation is more likely to move samples across class-specific decision boundaries to generate much more confusing samples. In our opinion, this is a general phenomenon in pattern classification, not particular to adversarial attacks.

## 5 FINAL REMARKS

In this paper, we have presented some theoretical results by Fourier analysis of ReLU neural networks. These results are useful for us to understand why neural networks are vulnerable to adversarial samples. Based on the results, we hypothesize that the inevitable and ubiquitous existence of tiny unlearned regions in the model function mapping may be a major reason for adversarial vulnerability. As a possible defence strategy, we have proposed a simple post-averaging method. Experimental results on the ImageNet and the CIFAR-10 datasets have demonstrated that our simple defence technique turns out to be very effective against many popular attack methods in the literature. Finally, it will be interesting to see whether our post-averaging method will be still robust against any new attack methods in the future.

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

APPENDIX: MATHEMATICAL PROOFS

**Definition B.1.** *A piece-wise linear function is a continuous function $f : \mathbb{R}^n \to \mathbb{R}$ such that there are some hyperplanes passing through origin and dividing $\mathbb{R}^n$ into $M$ pairwise disjoint regions $\mathfrak{R}_m$, $(m = 1, 2, ..., M)$, on each of which $f$ is linear:*

$$f(\mathbf{x}) = \begin{cases} \mathbf{w}_1 \cdot \mathbf{x} & \mathbf{x} \in \mathfrak{R}_1 \\ \mathbf{w}_2 \cdot \mathbf{x} & \mathbf{x} \in \mathfrak{R}_2 \\ \vdots \\ \mathbf{w}_M \cdot \mathbf{x} & \mathbf{x} \in \mathfrak{R}_M \end{cases}$$

**Lemma B.2.** *Composition of a piece-wise linear function with a ReLU activation function is also a piece-wise linear function.*

*Proof.* Let r(.) denote the ReLU activation function. If $f(\mathbf{x})$ on region $\mathfrak{R}_m$ takes both positive and negative values, $r\big(f(\mathbf{x})\big)$ will break it into two regions $\mathfrak{R}_p^+$ and $\mathfrak{R}_p^0$. On the former $r\big(f(\mathbf{x})\big) = f(\mathbf{x})$ and on the latter $r\big(f(\mathbf{x})\big) = 0$, which both are linear functions. As $f(\mathbf{x})$ on $\mathfrak{R}_p$ is linear, common boundary of $\mathfrak{R}_p^+$ and $\mathfrak{R}_p^0$ lies inside a hyperplane passing through origin – which is the kernel of the linear function. Therefore, if $f(\mathbf{x})$ is a piece-wise linear function defined by $k$ hyperplanes resulting in $M$ regions, $r\big(f(\mathbf{x})\big)$ will be a piece-wise linear function defined by at most $k+m$ hyperplanes. $\quad\square$

**Theorem B.3.** *The output of any hidden unit in an unbiased fully-connected ReLU neural network is a piece-wise linear function.*

*Proof.* This proposition immediately follows lemma B.2. $\quad\square$

**Definition B.4.** *Let $\mathbf{V} = \{\mathbf{v}_1, \mathbf{v}_2, ..., \mathbf{v}_n\}$ be a set of $n$ independent vectors in $\mathbb{R}^n$. An infinite simplex, $\mathfrak{R}_{\mathbf{V}}^+$, is defined as the region linearly spanned by $\mathbf{V}$ using only positive weights:*

$$\mathfrak{R}_{\mathbf{V}}^+ = \{\sum_{k=1}^n \alpha_k \mathbf{v}_k \mid \forall k \, \alpha_k > 0\} \tag{9}$$

**Theorem B.5.** *Each piece-wise linear function $f(\mathbf{x})$ can be formulated as a summation of some functions: $f(\mathbf{x}) = \sum_{k=1}^K f_k(\mathbf{x})$, each of which is linear and non-zero only in an infinite simplex as follows:*

$$f_k(\mathbf{x}) = \begin{cases} \mathbf{w}_k \cdot \mathbf{x} & \mathbf{x} \in \mathfrak{R}_{\mathbf{V}_k}^+ \\ 0 & otherwise \end{cases}$$

*where $\mathbf{V}_k$ is a set of $n$ independent vectors, and $\mathbf{w}_k$ is a weight vector.*

*Proof.* Each region $\mathfrak{R}_p$ of a piece-wise linear function, $f(\mathbf{x})$, which describes the behavior of a ReLU node if intersects with an affine hyper-plane results in a convex polytope. This convex polytope can be triangulated into some simplices. Define $\mathbf{V}_k$, $(k = 1, 2, ..., K)$, sets of vertexes of these simplices. The infinite simplexes created by these vector sets will have the desired property and $f(\mathbf{x})$ can be written as: $f(\mathbf{x}) = \sum_{k=1}^K f_k(\mathbf{x})$. $\quad\square$

As explained earlier in the original article by adding $n^2$ hyper-planes to those defining the piece-wise linear function, the output of a ReLU node may be represented as $f(\mathbf{x}) = \sum_{q=1}^Q g_q(\mathbf{x})$. These hyper-planes are those perpendicular to standard basis vectors and subtraction of one of these vectors from another one. That is, $\mathbf{e}_i$ $(i = 1, \ldots, n)$ and $\mathbf{e}_i - \mathbf{e}_j$ $(1 \le i < j \le n)$. Given this representation, the final step to achieve the Fourier transform is the following lemma:

**Lemma B.6.** *Fourier transform of the following function:*

$$s(\mathbf{x}) = \begin{cases} h(1 - \mathbf{1} \cdot \mathbf{x}) & \mathbf{x} \in \mathfrak{R}_{\mathbf{V}_*}^+ \\ 0 & otherwise \end{cases}$$

*may be presented as:*

$$S(\boldsymbol{\omega}) = \left(\frac{-i}{\sqrt{2\pi}}\right)^n \sum_{r=0}^{n} \frac{e^{-i\omega_r}}{\prod_{r'\neq r}(\omega_{r'} - \omega_r)} \tag{10}$$

*where $\omega_r$ is the $r$th component of frequency vector $\boldsymbol{\omega}$ $(r = 1, \cdots, n)$, and $\omega_0 = 0$.*

*Proof.* Alternatively, $s(\mathbf{x})$ may be represented as:

$$s(\mathbf{x}) = h(\mathbf{1} \cdot \mathbf{x})h(1 - \mathbf{1} \cdot \mathbf{x}) \prod_{j=1}^{n} h(x_j)h(1 - x_j) \tag{11}$$

Therefore, we need to compute Fourier transform of $h(x)h(1 - x)$:

$$\frac{1}{\sqrt{2\pi}} \int_{-\infty}^{\infty} e^{-ix\omega} h(x)(1 - x)\mathrm{d}x = \frac{1}{\sqrt{2\pi}} \int_{0}^{1} e^{-ix\omega}\mathrm{d}x \tag{12}$$

$$= \frac{-i}{\sqrt{2\pi}} \frac{1 - e^{-i\omega}}{\omega} \tag{13}$$

By taking the inverse Fourier transform of the function:

$$(\sqrt{2\pi})^{n-1} \int_{-\infty}^{\infty} \frac{-i}{\sqrt{2\pi}} \frac{1 - e^{-i\zeta}}{\zeta} \boldsymbol{\delta}_n(\boldsymbol{\omega} - \zeta\mathbf{1}) \, \mathrm{d}\zeta \tag{14}$$

where $\boldsymbol{\delta}_n$ is $n$-dimensional Dirac Delta function, it can be shown that it is the Fourier transform of $h(\mathbf{1} \cdot \mathbf{x})h(1 - \mathbf{1} \cdot \mathbf{x})$:

$$\left(\frac{1}{\sqrt{2\pi}}\right)^n \int \cdots \int_{\mathbb{R}^n} e^{i\boldsymbol{\omega}.\mathbf{x}} (\sqrt{2\pi})^{n-1} \int_{-\infty}^{\infty} \frac{-i}{\sqrt{2\pi}} \frac{1 - e^{-i\zeta}}{\zeta} \boldsymbol{\delta}_n(\boldsymbol{\omega} - \zeta\mathbf{1}) \, \mathrm{d}\zeta \, \mathrm{d}\boldsymbol{\omega} \tag{15}$$

$$= \frac{1}{\sqrt{2\pi}} \int \cdots \int_{\mathbb{R}^n} e^{i\boldsymbol{\omega}.\mathbf{x}} \int_{-\infty}^{\infty} \frac{-i}{\sqrt{2\pi}} \frac{1 - e^{-i\zeta}}{\zeta} \boldsymbol{\delta}_n(\boldsymbol{\omega} - \zeta\mathbf{1}) \, \mathrm{d}\zeta \, \mathrm{d}\boldsymbol{\omega} \tag{16}$$

$$= \frac{1}{\sqrt{2\pi}} \int_{-\infty}^{\infty} \frac{-i}{\sqrt{2\pi}} \frac{1 - e^{-i\zeta}}{\zeta} \int \cdots \int_{\mathbb{R}^n} e^{i\boldsymbol{\omega}.\mathbf{x}} \boldsymbol{\delta}_n(\boldsymbol{\omega} - \zeta\mathbf{1}) \, \mathrm{d}\boldsymbol{\omega} \, \mathrm{d}\zeta \tag{17}$$

$$= \frac{1}{\sqrt{2\pi}} \int_{-\infty}^{\infty} \frac{-i}{\sqrt{2\pi}} \frac{1 - e^{-i\zeta}}{\zeta} e^{i\zeta\mathbf{1}.\mathbf{x}} \, \mathrm{d}\zeta \tag{18}$$

$$= h(\mathbf{1} \cdot \mathbf{x})h(1 - \mathbf{1} \cdot \mathbf{x}) \tag{19}$$

Now we can find the Fourier transform of $s(\mathbf{x})$

$$S(\boldsymbol{\omega}) = \left(\prod_{r=1}^{n} \frac{-i}{\sqrt{2\pi}} \frac{1 - e^{-i\omega_r}}{\omega_r}\right) * (\sqrt{2\pi})^{n-1} \int_{-\infty}^{\infty} \frac{-i}{\sqrt{2\pi}} \frac{1 - e^{-i\zeta}}{\zeta} \boldsymbol{\delta}_n(\boldsymbol{\omega} - \zeta\mathbf{1}) \, \mathrm{d}\zeta \tag{20}$$

$$= i\left(\frac{-i}{\sqrt{2\pi}}\right)^{n+2} \int_{-\infty}^{\infty} e^{-i\zeta} \prod_{r=0}^{n} \frac{1 - e^{-i(\omega_r-\zeta)}}{\omega_r - \zeta} \, \mathrm{d}\zeta \tag{21}$$

where $*$ is convolution operator. The final integrand may be represented as:

$$e^{-i\zeta} \prod_{r=0}^{n} \frac{1 - e^{-i(\omega_r-\zeta)}}{\omega_r - \zeta} = e^{-i\zeta} \prod_{r=0}^{n} \frac{1}{\omega_r - \zeta} \prod_{r=0}^{n}(1 - e^{-i(\omega_r-\zeta)}) \tag{22}$$

$$= e^{-i\zeta} \sum_{r=0}^{n} \frac{A_r}{\omega_r - \zeta} \prod_{r=0}^{n}(1 - e^{-i(\omega_r-\zeta)}) \tag{23}$$

$$= e^{-i\zeta} \sum_{r=0}^{n} \frac{A_r}{\omega_r - \zeta} \sum_{B \subseteq \Omega}(-1)^{|B|} e^{-i(\sigma_B - |B|\zeta)} \tag{24}$$

$$= \sum_{r=0}^{n} \frac{A_r}{\omega_r - \zeta} \sum_{B \subseteq \Omega}(-1)^{|B|} e^{-i(\sigma_B - (|B|-1)\zeta)} \tag{25}$$

where $\Omega = \{\omega_0, ..., \omega_n\}$, $\sigma_B$ is the summation over elements of $B$ and $A_r = \prod_{r' \neq r} \frac{1}{\omega_{r'} - \omega_r}$. Therefore:

$$\int_{-\infty}^{\infty} e^{-i\zeta} \prod_{r=0}^{n} \frac{1 - e^{-i(\omega_r - \zeta)}}{\omega_r - \zeta} \, d\zeta \tag{26}$$

$$= \int_{-\infty}^{\infty} \sum_{r=0}^{n} \frac{A_r}{\omega_r - \zeta} \sum_{B \subseteq S} (-1)^{|B|} e^{-i(\sigma_B - (|B|-1)\zeta)} \, d\zeta \tag{27}$$

$$= \sum_{r=0}^{n} A_r \int_{-\infty}^{\infty} \frac{1}{\omega_r - \zeta} \sum_{B \subseteq S} (-1)^{|B|} e^{-i(\sigma_B - (|B|-1)\zeta)} \, d\zeta \tag{28}$$

$$= \sum_{r=0}^{n} A_r \int_{-\infty}^{\infty} \frac{1}{\zeta} \sum_{B \subseteq S} (-1)^{|B|+1} e^{-i(\sigma_B - (|B|-1)\omega_r + (|B|-1)\zeta)} \, d\zeta \tag{29}$$

$$= \sum_{r=0}^{n} A_r \sum_{B \subseteq S} (-1)^{|B|} i\pi \, \text{sign}(|B| - 1) e^{-i(\sigma_B - (|B|-1)\omega_r)} \tag{30}$$

If $B$ does not contain $\omega_r$ and have at least 2 elements then the terms for $B$ and $B \cup \{\omega_r\}$ will cancel each other out. Also, $\text{sign}(|B| - 1)$ will vanish if $B$ has only one element. Therefore, there only remains empty set and sets with two elements one of them being $\omega_r$. Given the fact that $\sum A_r = 0$, the result of the integral will be:

$$\int_{-\infty}^{\infty} e^{-i\zeta} \prod_{r=0}^{n} \frac{1 - e^{-i(\omega_r - \zeta)}}{\omega_r - \zeta} \, d\zeta = i\pi \sum_{r=0}^{n} A_r \left( -e^{-i\omega_r} + \sum_{r' \neq r} e^{-i\omega_{r'}} \right) \tag{31}$$

$$= -2i\pi \sum_{r=0}^{n} A_r e^{-i\omega_r} \tag{32}$$

Finally, substituting 32 into 21 yields to the desired result. $\qquad \square$

**Theorem B.7.** *The Fourier transform of the output of any hidden node in a fully-connected ReLU neural network may be represented as $\sum_{q=1}^{Q} \mathbf{w}_q \mathbf{A}_q^{-1} \nabla S(\boldsymbol{\omega} \mathbf{A}_q^{-1})$, where $\nabla$ denote the differential operator.*

*Proof.* As discussed in the original paper, $f(\mathbf{x}) = \sum_{q=1}^{Q} g_q(\mathbf{x})$ where:

$$g_q(\mathbf{x}) = \begin{cases} \bar{\mathbf{w}}_q \cdot \bar{\mathbf{x}}_q \, h(1 - \mathbf{1} \cdot \bar{\mathbf{x}}_q) & \bar{\mathbf{x}}_q \in \mathfrak{R}_{\mathbf{V}_*}^+ \\ 0 & \text{otherwise} \end{cases} \tag{33}$$

or equivalently:

$$g_q(\mathbf{x}) = \bar{\mathbf{w}}_q \cdot \bar{\mathbf{x}}_q s(\bar{\mathbf{x}}_q) \tag{34}$$

Therefore:

$$F(\boldsymbol{\omega}) = \sum_{q=1}^{Q} G_q(\boldsymbol{\omega}) \tag{35}$$

$$= \sum_{q=1}^{Q} \bar{\mathbf{w}}_q . \nabla S(\bar{\boldsymbol{\omega}}_q) \tag{36}$$

where $\bar{\boldsymbol{\omega}}_q = \boldsymbol{\omega} \mathbf{A}_q^{-1}$. $\qquad \square$

Derivation of eq.(6)

As for the Fourier transform computed in section 3.1, it should be mentioned that the integral in equation 6 is the Fourier transform of:

$$h_r(\mathbf{x}) = h(r - |\mathbf{x}|) \tag{37}$$

which can be derived utilizing the property of the Fourier transforms for radially symmetric functions (Stein and Weiss, 1971):

$$H_r(\boldsymbol{\omega}) = |\boldsymbol{\omega}|^{-\frac{n-2}{2}} \int_0^\infty J_{\frac{n-2}{2}}(|\boldsymbol{\omega}|\rho)\rho^{\frac{n-2}{2}} h(r-\rho)\rho \, \mathrm{d}\rho \tag{38}$$

$$= |\boldsymbol{\omega}|^{-\frac{n-2}{2}} \int_0^r J_{\frac{n-2}{2}}(|\boldsymbol{\omega}|\rho)\rho^{\frac{n}{2}} \, \mathrm{d}\rho \tag{39}$$

$$= (\frac{r}{|\boldsymbol{\omega}|})^{\frac{n}{2}} J_{\frac{n}{2}}(r|\boldsymbol{\omega}|) \tag{40}$$

Given this transform:

$$F_C(\boldsymbol{\omega}) = F(\boldsymbol{\omega})\frac{1}{\mathbf{V}_C} \int \cdots \int_{\mathbf{x}' \in C} e^{-i\mathbf{x}' \cdot \boldsymbol{\omega}} \, \mathrm{d}\mathbf{x}' \tag{41}$$

$$= F(\boldsymbol{\omega})\frac{\Gamma(\frac{n}{2}+1)}{\pi^{\frac{n}{2}}} \frac{J_{\frac{n}{2}}(r|\boldsymbol{\omega}|)}{(r|\boldsymbol{\omega}|)^{\frac{n}{2}}} \tag{42}$$

