# OpenReview forum: "Bandlimiting Neural Networks Against Adversarial Attacks"
_ICLR.cc/2020/Conference — Reject_

### Official Review · AnonReviewer2 · 2019-10-19
**Official Blind Review #2**

**Rating:** 1

**Review:**

The paper makes three contributions: 1) analyzing the number of linear regions and the Fourier spectrum of fully-connected neural networks with ReLU activations; 2) proposing an ensemble method as a defense against adversarial evasion attacks; 3) analyzing the performance of that defense against state-of-the-art attacks on CIFAR-10 and ImageNet data.

My main concern is regarding the originality and significance of this work. Most of the results regarding the piece-wise linear geometry of fully-connected neural networks with ReLU activations are known. The authors should reference and discuss the relation of their work in particular to [1]. The Fourier transformation results look straightforward to me. I am not convinced either that they are indeed "essential", as the authors write, for analyzing the nature of adversarial samples. In fact, I find the discussion in Section 2.2 rather vague; for instance, what is the formal notion of "tiny unlearned regions" of neural networks? The authors mention an experiment in which they observed that an adversarial perturbation on ImageNet led to the crossing of N=5278 hyperplanes "per layer", which sounds like an interesting finding, however, more detail is needed to understand how it was obtained.

The proposed defense is not very original either; the authors should discuss it in comparison e.g. to [2] and [3] who also considered ensemble-based methods to defend against adversarial perturbations. The experimental results in Table 1 & 2 look promising at a first glance, however, [4] has demonstrated that ensemble-based, randomized defenses can be fairly easily defeated (e.g. using expectations over transformations when computing the gradients for adversarial sample generation). The authors need to report the performance of their defense under such white-box settings.

The strong performance of the FGSM attack in Figure 3 is surprising. PGD is strictly stronger than FGSM, so its defense rate shouldn't be higher than the one for FGSM. I disagree with the authors' explanation that FGSM "generates much larger perturbations"; if the epsilon parameter is set properly in the PGD attack, it should exhaust that perturbation budget (assuming that the step size and number of iterations are large enough). On the other hand, I don't understand how the authors were able to constrain the L-inf norm of the C&W and DeepFool adversarial samples (since those attacks aim to minimize L2 norms).


[1] Montufar et al., On the Number of Linear Regions of Deep Neural Networks. NeurIPS, 2014.
[2] Liu et al., Towards robust neural networks via random self-ensemble. ECCV, 2018.
[3] Cao & Gong, Mitigating evasion attacks to deep neural networks via region-based classification. ACSAC, 2017.
[4] Athalye et al., Obfuscated gradients give a false sense of security. ICML, 2018.


**Experience Assessment:**

I have published one or two papers in this area.

**Review Assessment: Checking Correctness Of Derivations And Theory:**

I did not assess the derivations or theory.

**Review Assessment: Checking Correctness Of Experiments:**

I assessed the sensibility of the experiments.

**Review Assessment: Thoroughness In Paper Reading:**

I read the paper at least twice and used my best judgement in assessing the paper.

---

### Official Review · AnonReviewer3 · 2019-10-24
**Official Blind Review #3**

**Rating:** 3

**Review:**

Overview:
The paper is dedicated to studying adversarial attack and defense problems from the perspective of Fourier analysis. They demonstrate that the adversarial vulnerability of neural networks can be attributed to non-zero high-frequency components. Then, the author proposes a simple post average approach to smooth out the insignificant high-frequency components, which can improve the adversarial robustness of neural networks. They conduct extensive experiments on ImageNet and CIFAR-10 to defend existing attacks, including FGSM, PGD, DeepFool, and C&W attacks.

Strength Bullets:
1. The logic chain of this paper is complete. The author first gives an interesting observation that the non-zero high-frequency components may cause the vulnerability of neural networks. Then they propose a post average technique to reduce the high-frequency components and improve robustness.
2. The paper is well organized. Some figures, like fig1, are intuitive and interesting.

Weakness Bullets:
1. How can we get formula (4)? The author needs to provide more explanation.
2. It is confused about how to pick directional vectors v for input x. Does the author treat the input RGB image (i.e. 3 times 32 times 32) as a one-dimensional vector? Then the author chooses k other vectors in the neighborhood. There need more details and rational explanations. The author can use the visualization method (i.e. tSNE) the illustrate the relationship among samples.
3. For different sampling strategies, the author needs to provide an ablation study.
4. The most arguable point is the experiment setting. I think it is a weak white-box experiment setting. The author uses the attacked set which is generated from the undefended model. It means the author doesn't attack his post-processing method. Thus, the improved robustness can be the result of gradient masking introduced by the author's techniques. BTW, it is totally traceable to attack the paper's approach. We can formula equation (7) and equation (8) as two layers in the front and at the end of networks. Then you can calculate the gradient for a complete model setting. The author needs to provides more convincing results.
5. The author needs to compare with other post- or pre- processing methods, even the normal adversarial training. Also, the paper should contain detailed settings about attacks. For example, how many iterations do you run for PGD attacks? This is an important factor for trustworthy results.
6. [Minor] Quotation mask mistake in the second row of page 2.

Recommendation:
Due to the limits of experiments design and setting, this is a weak reject.

**Experience Assessment:**

I have published one or two papers in this area.

**Review Assessment: Checking Correctness Of Derivations And Theory:**

I carefully checked the derivations and theory.

**Review Assessment: Checking Correctness Of Experiments:**

I carefully checked the experiments.

**Review Assessment: Thoroughness In Paper Reading:**

I read the paper thoroughly.

---

### Official Review · AnonReviewer1 · 2019-10-24
**Official Blind Review #1**

**Rating:** 6

**Review:**

The paper proposes an approach for improving robustness of already trained artificial neural networks with relu activation functions. The main motivation comes from signal processing where robustness is typically obtained via averaging moduli of Fourier coefficients over some frequency band (e.g., mel-frequency coefficients and deep scattering spectrum are based on this principle). The strategy amounts to sampling several random direction vectors in a ball of constant radius centered at a training example and averaging their predictions. The empirical estimate of the expected predictor value over the ball centered at a training example is used as its hypothesis value.

The approach is introduced by first expressing an artificial neural network with relu activations as a piecewise linear function (Definition 2.1, Lemma 2.2, Theorem 2.3). The paper then makes an observation that the instance space can be further sub-divided such that the network can be written as a sum of piecewise functions defined over regions given by positive linear combinations of linearly independent vectors (Definition 2.4, Theorem 2.5). A linear transformation of the instance space then allows for writing that function in canonical basis and computing its Fourier transform (Lemma 2.6, Theorem 2.7). The paper then makes an observation that the inverse of the linear transform used for making a change of basis (mapping to the canonical basis) can introduce instabilities to piecewise linear components defined over small regions (the inverse matrix appears in the Fourier transform). To address this instability, the paper relies on prior work by Jiang et al. (1999, 2003) and assigns the expected value over some ball centered at a training example as its prediction (which should be equivalent to performing averaging over bands in the Fourier domain). The integral is analytically intractable and, thus, the authors do an empirical estimate by averaging values in different random directions around the example. The experiments show that the method does not exhibit any serious instability with respect to the number points selected in that way.

The strategy does not require any additional training of the network and can be easily applied to already trained models with relu activations.

In the experiments, the approach is evaluated on standard adversarial attacking mechanisms: fast gradient sign method (Goodfellow et al., 2014), projected gradient descent method (Kurakin et al., 2016; Madry et al., 2017), deep fool attack method (Moosavi-Dezfooli et al., 2016) and L_2 method (Carlini & Wagner, 2017). I am not very familiar with the related work but this seems to be sufficient number of baselines to assess the effectiveness of the approach. The experiments are performed on ImageNet and Cifar10 datasets and show that the approach can make standard ResNet models more robust to the listed attacking strategies. It might be useful to test the approach with several different architectures (e.g., multi-layer perceptrons with different number of hidden layers, mix of convolutional and fully connected blocks etc).

In summary, the paper is well written and easy follow. The idea itself is simple but the intuition behind it is rather interesting and (to the best of my knowledge) provides novel insights into the workings of artificial neural networks with relu activations.

**Experience Assessment:**

I do not know much about this area.

**Review Assessment: Checking Correctness Of Derivations And Theory:**

I assessed the sensibility of the derivations and theory.

**Review Assessment: Checking Correctness Of Experiments:**

I assessed the sensibility of the experiments.

**Review Assessment: Thoroughness In Paper Reading:**

I made a quick assessment of this paper.

---

### Decision · Program_Chairs · 2019-12-19

**Decision:**

Reject

**Comment:**

The reviewers recommend rejection due to various concerns about novelty and experimental validation. The authors have not provided a response.